# Characterizing Foot and Leg Scores for Montana’s Registered Angus Cattle

**DOI:** 10.3390/ani13182849

**Published:** 2023-09-07

**Authors:** Taylre Sitz, Hannah DelCurto-Wyffels, Megan Van Emon, Sam Wyffels, Jeremiah Peterson, Thomas Hamilton, Kelli Retallick, Esther Tarpoff, Andre Garcia, Kurt Kangas, Tim DelCurto

**Affiliations:** 1Department of Animal and Range Sciences, Montana State University, Bozeman, MT 59717, USA; taylreesitz@gmail.com (T.S.); hannah.delcurto@montana.edu (H.D.-W.); megan.vanemon@montana.edu (M.V.E.); samwyffels@montana.edu (S.W.); petersonjeremiah7@gmail.com (J.P.); thomas.hamilton2@montana.edu (T.H.); 2American Angus Association, Saint Joseph, MO 64506, USA; kretallick@angus.org (K.R.); etarpoff@angus.org (E.T.); agarcia@angus.org (A.G.); kkangas@angus.org (K.K.)

**Keywords:** beef cattle, foot angle, claw set, longevity, structural soundness

## Abstract

**Simple Summary:**

As the beef industry continues to become more efficient and profitable, structural soundness will be important for the longevity of forage-based beef cattle. This study characterized the effects of age, sex, and sire line on mean foot angle and claw set scores of registered Angus cattle in Montana, USA. In our study, yearling bulls had greater mean foot angle and claw set scores than yearling heifers. Likewise, a greater proportion of yearling bulls had foot scores that were not ideal compared to yearling heifers. In addition, mean foot angle and claw set scores increased with increasing cow age, as well as the proportion of cattle with nonideal foot scores. The location of the combined worst foot was also influenced by cow age, with yearling cattle having predominantly front foot issues whereas mature cows often experienced hind foot issues. These data could potentially be used to improve the current American Angus Association (AAA) models and protocols for foot scoring.

**Abstract:**

The objective of this study was to characterize foot angle and claw set scores of Montana’s (USA) registered Angus cattle using a total of 4723 cattle scored: 1475 yearling bulls, 992 yearling heifers, 1044 2- and 3-year-old cows, and 1212 cows ≥ 4 years old. Yearling bulls had a 0.12 and 0.20 greater mean foot angle and claw set score, respectively, compared to yearling heifers (*p* < 0.01). Foot angle and claw set scores increased (*p* < 0.01) with advancing cow age. The combined worst foot changed quadratically with age (*p* < 0.01) with the majority of problem feet in cows aged 2 to 3 years and older being hind feet issues. The proportion of foot angle and claw set scores not equal to 5 increased quadratically with age (*p* < 0.01), with heifers having the lowest proportion of scores not equal to 5 (15.8 and 31.7%, respectively) compared to cows aged 4 years and older. Sire lines had an effect on progeny claw set (*p* < 0.05) and foot angle scores (*p* < 0.05), as well as variation of progeny foot scores. These data could potentially be used to refine expected progeny difference models.

## 1. Introduction

Proper foot conformation is important for productivity, health, and welfare in beef cattle production systems [1,2]. Correct bovine foot structure consists of a claw set with symmetrical toes of appropriate length, and a foot angle of approximately 45 degrees that leaves an appropriate depth of heel [3]. Optimal locomotion occurs with proper conformation, as many anatomical parts must articulate together for movement [1,2]. Nutrition and breeding are crucially affected by structural soundness and free locomotion, as both are important for animals to cover acres of pasture [4]. Therefore, structural soundness is considered a key factor related to reproductive efficiency, cow weight, body condition maintenance, and calf growth [5]. Proper conformation is associated with longevity due to the locomotive requirements of cow-calf production in extensive forage-based systems [1,6,7]. Additionally, mobility and soundness are important for cattle regardless of the management system, from cattle grazing extensive rangelands to feedlot cattle [1,8]. Structural soundness supports animal health and well-being [1]. Specifically, forage-based cattle systems require increased locomotion expectations for animals to find feed, water, salt, and minerals [9,10]. Extensive rangeland systems encompass a significant portion of beef production in the Western United States and management for these systems often aims to maximize animal performance by enhancing the intake and digestion of forage resources [10,11]. The enhancement of intake and digestion is achieved by the ability of animals to traverse greater distances and utilize more challenging topography [12]. In respect to the cattle feeding industry, selection and management pressure has resulted in substantial increases in carcass weight [13]. As a result, as feeder cattle growth and performance increase, proper foot and leg structure will continue to be an important consideration for selection [1].

Many factors likely affect foot conformation such as current production expectations, size and stature, age [14,15,16], nutrition [17,18], infection or trauma [19], trauma [20], genetics [21,22], environment [23], and management [23,24,25]. While numerous environmental factors have been suggested to affect foot conformation and health, limited information is available [1,23]. Additionally, while research has evaluated lameness and foot issues in the dairy breeds, limited work has been conducted specific to beef cattle [3,6,8]. Increased knowledge and research as to more specifically how various factors affect foot conformation may allow us to make greater improvements in beef cattle.

In response to concerns regarding feet and leg conformation, the AAA foot angle and claw set expected progeny differences (EPD) were developed in 2019 [24]. These are relatively new EPDs; the current genetic evaluation at the AAA for foot angle and claw set have more than 180,000 records for each of these traits, in a joint genetic evaluation with the Canadian and Australian Angus Associations. With foot angle and claw set being newer traits to submit phenotypic scores, the amount of data is still relatively small compared with other traits such as birth and weaning weights (9.8 M and 10 M records, respectively), and continued data recording is essential to better characterize the population. Therefore, the objectives of this study were: (1) to increase the amount of phenotypic data available for enhancing the foot angle and claw EPDs of Montana genetics and (2) to evaluate factors that could impact foot angle and claw set scores. Specifically, this study evaluated the interaction of sex and age on the claw set and foot angle scores of front or hind legs. In addition, this study characterized the claw set and foot angle scores for Montana’s most commonly used sire lines in this dataset.

## 2. Materials and Methods

From November 2021 to April 2022 (5 months), we used the AAA Foot Scoring Guidelines to subjectively score claw set and foot angle on 12 Montana registered Angus herds of various sizes [1,26]. Over this time, 1475 yearling bulls, 992 yearling heifers, 1044 2- and 3-year-old cows, and 1212 cows aged 4 years and older were scored by two individuals. To limit bias among scorers, one scorer would score each herd. Scoring technicians were trained utilizing various video and print resources from the AAA, as well as scoring a series of animals together to calibrate individual scoring to a relative standard. This research project was a collaboration between the researchers and Montana Angus producers who, during normally scheduled cattle working events such as pregnancy checking, scour-guarding, or recording yearling weights, allowed the foot scores to be measured on their registered Angus cattle. The ranchers that volunteered for the project were contacted prior to the scoring of their cattle and asked to provide several points of data including the AAA registration number, tag and tattoo number, sex, birthdate and sire of the animals in the study. Scoring was conducted using AAA foot scoring guidelines [26] to determine and score the single worst foot for both traits. Individual scores for claw set and foot angle were recorded, as well as which foot was chosen as the combined worst foot. Upon completion of the scoring, data were compiled for ranchers to submit to the AAA at their discretion and utilize when making decisions on their farm and ranch.

The AAA foot scoring guideline system [26] uses a nine-point scale to visually appraise two traits: claw set and foot angle [Figure 1]. Scoring is performed for both traits on the animal’s single worst foot and utilizes only whole numbers. The ideal foot angle is a 45-degree angle at the pastern joint with appropriate toe length and heel depth (score of 5). Lower scores identify animals with feet that become straighter (>45° angle) or post legged while higher scores identify animals with a foot set too far under themselves (<45° angle) or a weaker pastern. The ideal claw set has symmetrical toes in length that are appropriately spaced (score of 5). A lower score equates to animals with severely divergent toes, whereas a higher score equates to animals with scissor-type toes or toes that are curled over themselves.

The AAA foot angle and claw set expected progeny differences (EPD) were developed in 2019 [15]. Currently, the selection tools utilize the 5–9 score range for both the foot angle and claw set (there are too few scores in the 1–4 range to be included in the selection tools). As of 2022, the AAA recommends foot scoring animals annually. Cattle should be over 320 days of age when scored for the first time, and all scoring should take place on solid, flat ground where the phenotypes can be most easily assessed. The foot angle EPD, as defined by the AAA, is expressed in units of foot angle score, with a lower EPD being more favorable, indicating a sire will produce progeny with a more ideal foot angle. The claw set EPD, as defined by the AAA, is expressed in units of claw set score, with a lower EPD being more favorable, indicating a sire will produce progeny with a more ideal claw set. 

Due to the relative newness of collecting claw set and foot angle scores in Angus cattle, a primary goal of this research project was to characterize foot angle and claw set scores of registered Angus cattle in Montana. All cattle scored in this study were classified for age as either yearling bulls, yearling heifers, 2- to 3-year-old cows, or cows aged 4 years and older to evaluate the effects of sex and age on foot angle and claw set scores (yearling heifers vs. yearling bulls; yearling vs. 2- to 3-year-old vs. ≥4-year-old cows). Foot angle and claw set data were then analyzed using generalized linear models in an ANOVA framework including either sex or cow age classification as a fixed effect. For cattle that had foot angle and/or claw set scores differing from the ideal score of 5, we evaluated whether sex or cow age influenced the probability of the combined worst foot being a front or hind foot issue using generalized linear models following a binomial distribution in an ANOVA framework, with either sex or cow age classification as a fixed effect. Preplanned orthogonal polynomial contrasts were used to determine linear and quadratic effects of cow age classification on foot angle and claw set scores. Additionally, all sire lines within the dataset that had ≥75 offspring scored (9 total) were used to characterize the effect of sire line on progeny foot angle and claw set scores. Progeny foot angle and claw set data were analyzed using generalized linear models in an ANOVA framework including sire line as a fixed effect. All data were plotted and natural log-transformed if needed to satisfy assumptions of normality and homogeneity of variance. Significance was determined at an alpha of <0.05 with a tendency discussed with alpha <0.10. All statistical analyses were performed in R [27].

## 3. Results

### 3.1. Yearling Bulls vs. Yearling Heifers

Claw set and foot angle records were analyzed on 1475 yearling bulls and 992 yearling heifers from Montana registered Angus producers (Table 1). Yearling bulls had a 0.12 greater mean foot angle score compared to yearling heifers (*p* < 0.01). Yearling bulls had 1.74 times the number of individuals with a foot angle score different from 5 compared to yearling heifers (*p* < 0.01; 27.5% vs. 15.8%, respectively). Yearling bulls and heifers were more likely to have front foot scores deviating from 5; however, yearling bulls had a greater proportion of front foot scores deviating from 5 compared yearling heifers (*p* < 0.01). Yearling bulls had a mean claw set score that was 0.2 points higher than yearling heifers (*p* < 0.01). Both yearling bulls and heifers were more inclined to have front foot scores that differed from 5. Yearling bulls exhibited a higher proportion of deviated claw set scores on the front foot compared to heifers (*p* < 0.01; Figure 2).

### 3.2. The Influence of Cow Age

Comparisons were made across three different age groups of female cattle–yearling heifers (n = 992), 2- and 3-year-old cows (n = 1044), and 4-year-old cows and older (n = 1212). Foot angle scores increased linearly (*p* < 0.01; Table 2) with advancing age, ranging from 5.15 to 5.80 for heifers and cows aged 4 years and older, respectively. In addition, the number of foot angle scores not equal to 5 increased quadratically with age (*p* < 0.01) with heifers being the lowest, cows aged 4 years and older being the highest, and 2- and 3-year-olds being intermediate. For cows that did not score a 5 for foot angle, the proportion of scores on the front versus hind foot was quadratically (*p* < 0.01) impacted by age class. Specifically, heifers had a higher proportion of front foot scores that deviated from 5. In contrast, 2- and 3-year-old cows and cows aged 4 years and older were observed to have a higher proportion of hind feet scores deviating from 5, with cows aged 4 years and older having the highest proportion of hind foot score deviation.

Claw set scores increased quadratically (*p* < 0.01; Table 2) with advancing age, ranging from 5.34 to 5.86 for heifers and cows aged 4 years and older, respectively. In addition, the number of claw set scores not equal to 5 increased quadratically with age (*p* < 0.01) with heifers being the lowest and cows aged 4 years and older being the highest. For cows not scoring a 5 for claw set, the proportion of scores on the front versus hind foot was quadratically (*p* < 0.01; Figure 3) impacted by age class. Specifically, heifers had a higher proportion of front foot scores that deviated from 5 versus hind foot scores. In contrast, 2- and 3-year-old cows and cows aged 4 years and older had a higher proportion of scores attributed to hind feet, with cows aged 4 years and older having a higher proportion of hind foot scores than the other two age classes.

### 3.3. Sire Line Progeny Analysis

A descriptive statistical analysis of the nine sires with at least 75 progeny was performed to determine the mean claw set and foot angle, the range of claw set and foot angle scores, and the coefficient of variance for each sire (Table 3, Figure 3). For foot angle, the range between the progeny of the sire line with the greatest foot angle score and that of the least was 0.60. For claw set, the same range was 0.57. Several differences were observed among sire lines in regard to the progeny mean foot scores. Specifically, sire 7 differs for foot angle from sires 1, 2, 3, and 4 (*p* < 0.05) and is similar to sires 5, 6, 8, and 9 (*p* < 0.05). Sire 4 had a greater foot angle than sires 1, 3, 5, 6, 7, 8, and 9 (*p* < 0.05). Sire 7 exhibited the most ideal mean foot angle score in his progeny with sire 4 exhibiting the least ideal mean foot angle scores. Sires 1, 2, and 4 had greater mean claw set scores in their progeny versus sires 3, 5, 7, 8, and 9 (*p* < 0.05). Generally, the coefficient of variation will decline with an increase in the number of foot score observations. Sire EPDs and the coefficient of variation of foot angle and claw set scores are listed in Table 3 for reference, but note that not all of the progeny records have been included in the AAA National Cattle Evaluation analysis. For the entire data set, foot angle and claw scores were moderately correlated (R = 0.34, *p* < 0.01; Figure 4), suggesting that as structural problems occur, they may influence both phenotypic traits.

## 4. Discussion

Changes in foot scores, in this study, show an effect on the claw set and foot angle scores associated with age, sex, and sire line. The relative difference in foot angle and claw set scores represent significant differences in foot and leg conformation. Most animals scored using the AAA guidelines fall into foot angle and claw set scores of 4–8, and the scoring system uses only whole numbers. Age at scoring is fit (co)variable and sex is part of the contemporary group definition for the genetic evaluation model used to predict foot angle and claw set EPDs [24]. Environment and subjectivity factors affect a score given to a particular animal on a particular day which is in part being addressed by the use of contemporary groups, subsets of animals of similar age, sex, and management type, at the AAA [28]. It is imperative that the same scorer scores all cattle throughout the scoring event. If more than one scorer is used throughout the day, this should be accounted for with additional contemporary groups associated with each additional scorer. To fully test this theory, an additional study would be warranted that includes two or more scorers evaluating the same set of cattle on the same day for direct comparison.

Advancing age influences foot angle and claw set. The effects of advancing age on foot conformation encompasses a variety of compounding stresses including pregnancies, lactations, wear and tear by everyday locomotion, and of course, the physiological effects of advancing age [11,25]. Furthermore, the pressure on the joints and feet of a 350 kg growing yearling is substantially different from that of a 600 kg mature cow [22]. As seen in the results, the foot angle increases linearly, which may indicate that the animal becomes weaker in their pastern due to a decreasing foot angle. Further studies may investigate the effects of age versus weight. Hahn et al. [11] similarly found that age is associated with a decrease in foot angle, particularly for the hind legs. Claw set increases quadratically indicating that as a cow ages, her claws become longer and more curled.

In addition, a significant transition of the scored foot from a front foot to a hind foot was observed with increased age. The analysis of which foot was scored when the score for each trait differed from score of 5 yielded a shift from the front foot to the hind foot, specifically in heifers in the 2- and 3-year-old age classes, bringing into question the most appropriate age at which female cattle should be scored. The question then becomes: do problem feet in yearlings translate to problem feet in older age cattle? Females of various age classes in this study were scored once. Perhaps future research could evaluate the same females year on year to observe whether the patterns reported in this study continue to exist, to further investigate the correlations and characterize the change from younger to older age cattle. The current recommendation of the AAA is to score registered female cattle annually [26].

Montana Angus producers register and sell a significant number of bulls: 18% of all registered Angus bulls sold in 2022 were from Montana [29]. Most data on bulls to date are limited to yearling age to 18-month-old bulls. While structure is important, longevity is not necessarily a driver of bull performance [29]. Since most bulls are sold by eighteen months of age, foot scoring bulls in purebred production systems may be confined to 12–18 months of age. We scored only bulls classifying as yearlings in this study. Additional research is needed to address the influence of age on foot angle and claw set scores in mature bulls.

Differences in management strategies and environment between the sexes could influence the claw set or foot angle. Physiological differences of bulls may result in increased body mass putting added pressure on front feet during the growing process. In contrast, the management of bulls versus heifers from weaning to yearling differs for most seedstock producers. Specifically, heifers are often managed very differently than their male counterparts, more often being allowed a slower and less intense growing period prepuberty. Heifers tend to more often be developed on pasture or in a feedlot, on a low-concentrate ration [30]. Many registered Angus bulls are marketed as yearlings, meaning they are expected to be well conditioned by that time and therefore spend much more time developing on higher energy rations [31,32]. These differences in management strategies between the sexes may attribute to the difference observed in this study.

The characterization of mean foot scores of progeny from sires with at least 75 offspring presented interesting and useful information for comparing sires. As presented in Table 3, the differences among mean foot angle and claw set scores are significant. The coefficient of variance for a sire’s scored offspring for the foot angle and claw set traits allow insight into the predictably of that sire’s progeny’s foot conformation. Sires with a smaller CV for progeny foot angle, for example, will generally produce offspring with foot scores that have less variability than sires with a larger CV. However, the age of progeny at the time of scoring may influence variability and, as a result, needs to be accounted for in genetic modeling.

For the entire data set, which included both mature and yearling female cattle, foot angle and claw scores were compared and found to be moderately correlated phenotypically (*R* = 0.34, *p* < 0.01). Wang [33] found the genetic correlation between these two traits to be 0.22 in yearling age cattle. A proposed causation for this relation is that the angle of the pastern/fetlock joint likely effects toe growth and wear. The AAA has found the heritability of foot angle and claw set scores to be low to moderate (0.16–0.37) [20,21]. This heritability range indicates that modest genetic progress may be made by breeding selection [33].

Our study will aid in the continued refinement of the foot scoring protocol. The frequency of hind versus front hoof selection is significantly influenced by age, with yearlings having predominantly front foot problems, whereas 2-year-old cows and older have predominantly hind foot problems. Additional improvements may be possible with continued model refinement and improvements with scoring guidelines specific to age and sex effects, such as enacting an age in which scoring is conducted or specifying whether a front or hind foot should be scored at a given age. The AAA guidelines have been beneficial in developing selection tools for optimal conformation. However, as with all tools, further refinement of models and sampling protocols are necessary to continue to grow the adoption of foot scoring protocols and genetic selection of beef cattle.

## 5. Conclusions

Beef breed associations and beef cattle breeders are emphasizing selection pressure on structural soundness. This increased need is likely due to the production demands placed on beef animals over centuries of cattle breeding. Ideal structural soundness often equates to better locomotion, which, in turn, increases an animal’s productivity and longevity in numerous production scenarios [1,5,6]. As the beef industry continues to become more efficient and profitable, structural soundness will continue to be important. The foot angle and claw set scoring guidelines are a start for creating a standard for measurement of foot quality in beef cattle. The foot scoring guidelines are based on visual appraisal of the foot angle and claw shape. There are numerous environmental and subjectivity factors that can affect a score given to a particular animal on a particular day, which are in part being addressed by the use of contemporary groups and by modeling the age in the genetic evaluation at the AAA.

Perhaps in the future, this subjective approach will lead to a more quantitative method with the use of technology (i.e., advanced imaging, pressure sensors, and activity monitors). Foot scoring guidelines provided by the AAA for their members allow the opportunity to continue to improve the selection tools for Angus cattle for foot angle and claw shape. The success of this EPD will depend upon acceptance by commercial cattlemen as a selection tool and the continued collection of foot scores by breeders.

## Figures and Tables

**Figure 1 animals-13-02849-f001:**
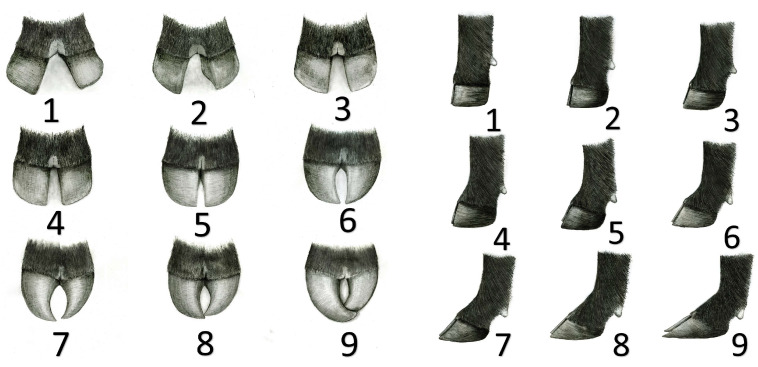
Foot score sketches courtesy of the American Angus Association characterizing claw scores (**left panels**) and pastern angles (**right panels**) [1].

**Figure 2 animals-13-02849-f002:**
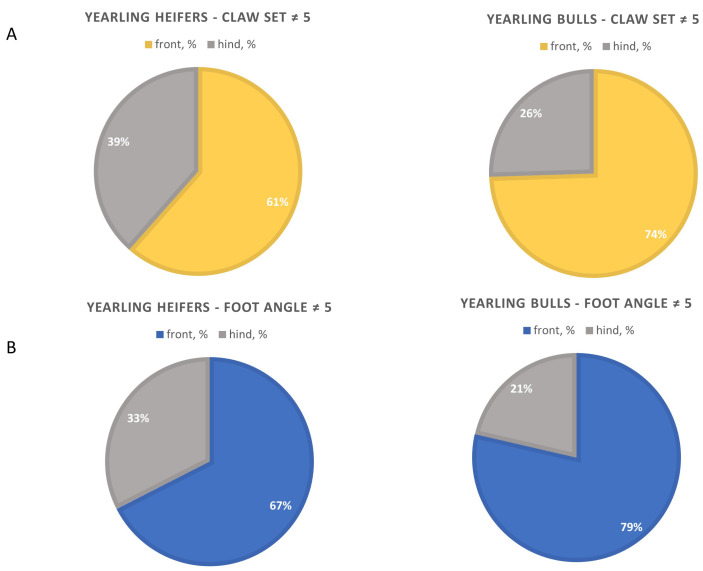
Proportion of Front Feet versus Hind Feet Scores ≠ 5 for Yearling Bulls and Heifers: Claw Set (**A**) and Foot Angle (**B**).

**Figure 3 animals-13-02849-f003:**
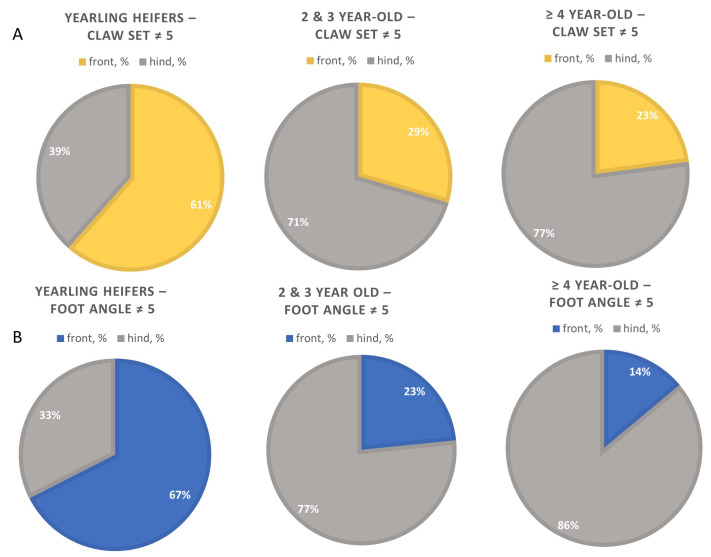
Influence of Cow Age on the Proportion of Front versus Hind Foot Issues for Claw Set (**A**) and Foot Angle (**B**) Scores ≠ 5.

**Figure 4 animals-13-02849-f004:**
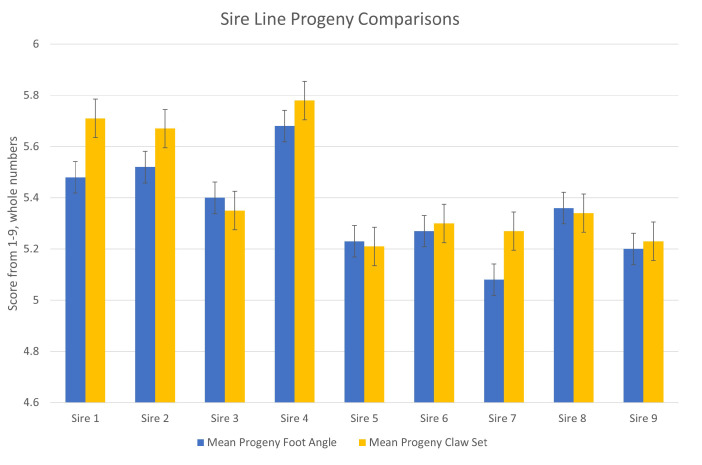
Mean Claw Set and Foot Angle Scores of Sires with ≥75 Progeny Scored. For entire data set, foot angle and claw scores were moderately correlated phenotypically (*R* = 0.34, *p* < 0.01).

**Table 1 animals-13-02849-t001:** Influence of sex on foot angle and claw set scores of yearling registered Angus cattle in Montana.

	Bulls	Heifers	SE ^1^	*p*-Value
Number of experimental units	1475	992		
Foot angle score ^2^	5.27	5.15	0.012	<0.01
Foot angle scores, % not 5 ^3^	27.5	15.8	1.2	<0.01
Front vs. hind ^4^			2.9	<0.01
Front, %	78.6	67.5		
Hind, %	21.4	32.5		
Claw set score ^2^	5.54	5.34	0.016	<0.01
Claw set scores, % not 5 ^3^	51.3	31.7	1.4	<0.01
Front vs. hind ^4^			2.2	<0.01
Front, %	74.5	61.5		
Hind, %	25.5	38.5		

^1^ Pooled standard error. ^2^ Scores based on the 1–9 scale used by the American Angus Association. ^3^ Percent of animals that did not score 5 (ideal score). ^4^ Data only from scores deviating from 5 performed as a binomial analysis.

**Table 2 animals-13-02849-t002:** Influence of age classification on foot angle and claw set scores of female registered Angus cattle in Montana.

					*p*-Values ^2^		
Age Class Comparison	Heifer	2 and 3 Year Old	≥4 Year Old	SE ^1^	Age	Linear	Quadratic
N ^3^	992	1044	1212				
Foot angle score ^4^	5.15	5.51	5.8	0.02	<0.01	<0.01	0.15
Foot angle score, % not 5 ^5^	15.8	45.1	66.0	1.4	<0.01	<0.01	<0.01
Claw set score ^4^	5.34	5.48	5.86	0.02	<0.01	<0.01	<0.01
Claw set score, % not 5 ^5^	31.7	42.8	68.2	1.4	<0.01	<0.01	<0.01

^1^ Pooled standard error. ^2^
*p*-values of age and preplanned orthogonal polynomial contrasts. ^3^ Number of experimental units. ^4^ Scores based on the 1–9 scale used by the American Angus Association. ^5^ Percent of animals that did not score 5 (ideal score) performed as a binomial analysis.

**Table 3 animals-13-02849-t003:** Influence of sire line on foot angle and claw set scores on registered Angus cattle in Montana.

	Sire										
	1	2	3	4	5	6	7	8	9	SE	*p*-Value
Number of offspring	77	220	96	95	123	84	80	80	75		
Average age of offspring	4.16 ^a^	2.50 ^b^	2.05 ^c^	3.16 ^d^	1.37 ^e^	1.44 ^e^	1.03 ^e^	1.34 ^e^	1.00 ^e^	0.12	<0.01
Offspring foot angle	5.48 ^abc^	5.52 ^ab^	5.4 ^ac^	5.68 ^b^	5.23 ^cd^	5.27 ^cd^	5.08 ^d^	5.36 ^acd^	5.2 ^cd^	0.06	<0.01
CV of offspring foot angle ^1^, %	10.09	12.06	11.29	12.38	8.05	8.51	6.4	10.38	8.94		
Sire foot angle EPD ^2^	+0.26	+0.47	+0.55	+0.41	+0.41	+0.37	+0.33	+0.29	+0.44		
EPD accuracy	0.78	0.86	0.80	0.78	0.86	0.71	0.81	0.75	0.63		
Offspring claw set	5.71 ^a^	5.67 ^a^	5.35 ^b^	5.78 ^a^	5.21 ^b^	5.30 ^b^	5.27 ^b^	5.34 ^b^	5.25 ^b^	0.07	<0.01
CV of offspring claw set ^3^, %	12.97	13.94	11.81	13.16	9.28	9.62	8.41	10.3	8.33		
Sire claw set EPD ^4^	+0.36	+0.36	+0.64	+0.36	+0.31	+0.27	+0.35	+0.50	+0.37		
EPD accuracy	0.78	0.86	0.80	0.78	0.87	0.71	0.81	0.75	0.63		

^1^ Coefficient of variation for foot angle expressed as a percent of the mean. ^2^ Expressed in units of foot angle score relative to ideal (5) with low value being more favorable. ^3^ Coefficients of variation for claw set expressed as a percent of the mean. ^4^ Expressed in units of claw set score relative to ideal (5) with low value being more favorable. ^a–e^ Variables within a row lacking a common superscript differ *p* < 0.05.

## Data Availability

Data are housed within the American Angus Association.

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
