# Peer review of "Characterizing Foot and Leg Scores for Montana’s Registered Angus Cattle"

_animals, 2023, doi:10.3390/ani13182849_

Round 1

Reviewer 1 Report

Enhancing claw health through selective breeding is undeniably essential. Utilizing the scoring systems outlined appears to be a promising approach in principle. However, it is crucial not to overlook the significant impact of management practices. The study's discussion highlights that differences in management strategies between males and females may have contributed to the observed variations (line 263-4: These differences in management strategies between the sexes may attribute to the difference observed in this study). Additionally, the subjectivity of scorers is acknowledged, potentially influencing the results.

Overall, the article presents a compelling examination of evaluation possibilities for beef cattle breeding. While not entirely groundbreaking, the topic is well-addressed, providing valuable insights. Thank you for sharing this interesting piece.

Author Response

These are the responses to all three reviewers.  In addition, reviewer #3 pdf is attached under reviewer #3.

Reviewer 2 Report

Intro

Line 38 cattle production systems - supporting reference needed.

Line 39 how long? there is primary literature that recommends toe lengths, although in this case, I feel that somewhere in the manuscript you need to include the illustrations that are in your reference [2] so that the reader can understand the current paper without having to go back and check the previous paper.

In the sentence beginning on line 41 to line 42 "nutrition and breeding are crucially affected by structural soundness and free locomotion" The meaning of this sentence is not entirely clear. Reading ref [2] helps. As above, I think maybe a little bit more information needs to be in this paper to allow the reader to understand it on its own without having to read ref [2] entirely to understand this paper…

Line 44 it would be nice to have more detail on what the ‘key factors related to performance’ actually are - rather than just skimming over them.

Later in line 44 you then say “proper conformation is associated with longevity due to these locomotive requirementsbut it is not 100% clear what these are.

Line 45 mobility and soundness are important for cattle regardless of management system - are there any implications for animal welfare?

Line 48 I struggled a little bit with the term "forage based cattle systems" because it's possible to be forage-based. Again, reading ref [2] made the context clearer, so I would try and briefly add the context into the introduction of this current paper to scaffold the reader. 

Line 50 you're aiming to “maximize animal performance”, but you haven't said what outcomes you're looking for with respect to evaluating performance. When you say “intake and digestion of food forage resources” are you referring to feed conversion efficiency or just literally digestion and utilization of nutrients?

Reference [2] information needs to be incorporated in pertinent places throughout this manuscript to scaffold the reader so they can understand the specific nomenclature around the system you are working in, and the scoring system that you're working to evaluate.

Line 57 you refer to limited work in beef cattle - it would be good to have some example references here, similarly some references as an example for the various factors and how they affect foot conformation and beef cattle later in the sentence.

Line 66 starts with the word weighs and I think this may be a typo should it be weights?

Line 67 to 69 The sentence beginning “Therefore the objectives of this study were…” This is a long, complicated sentence.  I think it just needs a bit of attention to make sure it has your intended meaning.

Line 72 right at the end of the sentence you say “in this data set” can these words be incorporated earlier in the sentence? They feel like an ‘add-on’ in their current position.

Materials and methods

Line 74 you've got a date range here, it might be worth putting brackets (x months) so that the reader hasn't actually got to sit there and work out how many months that was.

Line 76 how many farms/ herds were included in the study?

You include that farms were not split between the two individual scorers, but was there any attempt to look at the inter-scorer agreement? Did you train your scorers beforehand perhaps you could just enter that process into your materials and methods to inform the reader.

Line 86 I think it would be good to have an illustration of the scores that are being applied for claw set and foot angle appearances so that the reader really knows what it is that you're scoring and can think about what this means. Include a definition of the parameters evaluated so the method is repeatable.

Line 96 images would help immensely.

Lines 101 to 102 While I can follow the meaning of this sentence - this needs more context for the reader to follow smoothly e.g. the AAA currently uses... … to select animals (male and female) for breeding…

Lines 103 to 104As the AAA recommends foot scoring animals annually.” This feels like an incomplete sentence. The information that follows “Cattle should be over 320 days of age when scored for the first time and it should all take place in solid flat ground where phenotypes can be easily assessed. Feels like it would be better placed in the introduction rather than your methods.

Similarly, lines 111 to 113, would be better in your introduction laying the scene for the reader about what is currently done, the context of the study, and what its aims are.

Line 121...What is different about a 'GLM in an ANOVA framework'? I think you could justify the statistical methods used to your reader better.

Line 130 you talk about ‘a tendency’ - it would be preferable to not use this term and rather present your results with the spread and the P value that's associated with it for the reader to assimilate.

Results

Nice presentation with good legends.

Discussion

Line 223 'contemporary group build' What does this mean? I'm sure you know what you mean by this terminology, but for the first-time reader, this term needs to be explained further.

Line 230 Please provide a supporting reference for the pressure on the joints and feet being substantially different between the two sizes of animal.

Line 232 what do you mean by "weaker in their pasterns"? How can you measure this, and what does this literature say about weak pasterns?

Line 240age in which female cattle should be scored” I wonder if this is a typo and should read “age at which…”

Line 245… has there been any exploration of how foot angle and claw set scores relate to locomotion score or lameness?

Line 255 Do you mean differences in management strategies and environment?

Line 256-257 are there any supporting references looking at the physiological differences between bulls and heifers and how that affects the growing process and the weight distribution among the limbs?  

Similarly line 259-260 can you illustrate this sentence with rearing figures and supporting references.

Final paragraph of the discussion -  has anyone looked at the heritability of claw and foot conformation? Looking at reference [2] you have covered this in the previous paper. It might be worth linking to the previous paper or just including some of the pertinent figures in this conclusion.

How do these aspects of conformation relate to better outcomes for cattle with respect to locomotion? 

Conclusions

Line 291 “Beef breeds are emphasizing selection pressure on structural soundness.” the meaning of this statement is not entirely clear to the first-time reader. Do you mean the beef breed associations are looking to focus on structural soundness and good conformation? Can you reword it to explain your intended meaning to the reader?

Line 293 - 294 can you provide some evidence for this statement?

References

You have cited dairy papers that have looked at the scoring of foot and leg traits and evaluated heritability e.g. Hahn et al (1984). There are a number of other papers that could be worth consideration. To potentially provide more context and depth to the introduction and discussion of this manuscript:

Distl, O., Huber, M., Graf, F., Kräusslich, H., 1984. Claw measurements of young bulls at performance testing stations in Bavaria. Livestock Production Science 11, 587-598.

Politiek, R.D., Distl, O., Fjeldaas, T., Heeres, J., Mcdaniel, B.T., Nielsen, E., Peterse, D.J., Reurink, A., Strandberg, P., 1986. Importance of claw quality in cattle: review and recommendations to achieve genetic improvement. Report of the E.A.A.P. working group on "claw quality in cattle". Livestock Production Science 15, 133 - 152.

Distl, O., Koorn, D.S., Mcdaniel, B.T., Peterse, D., Politiek, R.D., Reurink, A., 1990. Claw traits in cattle breeding programs: Report of the E.A.A.P. working group "Claw quality in cattle". Livestock Production Science 25, 1-13.

Smit, H., Verbeek, B., Peterse, D.J., Jansen, J., McDaniel, B.T., Politiek, R.D., 1986a. Genetic aspects of claw disorders, claw measurements and 'type' scores for feet in Friesian cattle. Livestock Production Science 15, 205-217.

Boelling, D., Pollott, G.E., 1998a. Locomotion, lameness, hoof and leg traits in cattle I. Phenotypic influences and relationships. Livestock Production Science 54, 193-203. 

Boelling, D., Pollott, G.E., 1998b. Locomotion, lameness, hoof and leg traits in cattle II. Genetic relationships and breeding values. Livestock Production Science 54, 205-215.

Dal Zotto, R., De Marchi, M., Dalvit, C., Cassandro, M., Gallo, L., Carnier, P., Bittante, G., 2007. Heritabilities and genetic correlations of body condition score and calving interval with yield, somatic cell score, and linear type traits in brown swiss cattle. Journal of Dairy Science 90, 5737-5743.

Impact of age and sex – effects of parturition.

Offer, J.E., McNulty, D., Logue, D.N., 2000. Observations of lameness, hoof conformation and development of lesions in dairy cattle over four lactations. Veterinary Record 147, 105-109.

Tarlton, J.F., Holah, D.E., Evans, K.M., Jones, S., Pearson, G.R., Webster, J.F., 2002. Biomechanical and histopathological changes in the support structures of bovine hooves around the time of first calving. The Veterinary Journal 163, 196-204 

Knott, L., Tarlton, J.F., Craft, H., Webster, A.J.F., 2007. Effects of housing, parturition and diet change on the biochemistry and biomechanics of the support structures of the hoof of dairy heifers. The Veterinary Journal 174, 277-287.

Observer variation

Murray, R.D., 1994. Observer variation in field data describing foot shape in dairy cattle. Research in Veterinary Science 56, 265 - 269.

Calves  

Nüske, S., Scholz, A.M., Förster, M., 2003. Studies on the growth and the development of the Claw capsule in new born calves of different breeding lines using linear measurements. Archiv fur Tierzucht 46, 547-557.

Cattle Scottish highland

Nuss, K., Kolp,E.,  Braun, U., Weidmann, E., Haessig, M., 2013. Claw shape of Scottish Highland Cows after pasture and loose housing periods. 17th International Symposium and 9th International Conference on Lameness in Ruminants: Lameness in Ruminants: Past, Present and Future, 2013 -

Bristol, United Kingdom, 236-237.

Weight distribution and ground contact:

van der Tol, P.P.J., J.H.M. Metz, E.N. Noordhuizen-Stassen, W. Back, C.R. Braam, and W.A. Weijs. 2003. “The Vertical Ground Reaction Force and the Pressure Distribution on the Claws of Dairy Cows While Walking on a Flat Substrate.” Journal of Dairy Science 86 (9): 2875–83. doi:10.3168/jds.S0022-0302(03)73884-3.

Author Response

Thank you for the thoughtful and detailed review.  We appreciate your input and passion for the topic of feet and leg conformation.

Reviewer 3 Report

The topic of the study is very interesting and important for both the readers and the ranchers. Information about all factors that affect the final product is also important for producers and farmers. In addition, the foot angle and claw set scoring guidelines may be useful to other Angus breeder associations around the World.

All the reviwer's questions and recommendations can be found in the text.

Author Response

This is the edited version of reviewer #3.  We have responded to the questions and queries on the manuscript.  Most of the suggestions were incorporated into the manuscript.  Thank you for the thoughtful feedback.

Round 2

Reviewer 2 Report

.

Author Response

We have responded to reviewer 2 edits and questions on the manuscript.  Specifically, we have worked on reference citations and made additions as suggested by the reviewer.  Thank you for your thoughtful suggestions and edits.  We are confident that the revised manuscript is improved due to your suggestions.